# Chest Pain and Suspected Myocarditis Related to COVID-19 Vaccination in Adolescents—A Case Series

**DOI:** 10.3390/children9050693

**Published:** 2022-05-10

**Authors:** Da-Eun Roh, Hyejin Na, Jung-Eun Kwon, Insu Choi, Yeo-Hyang Kim, Hwa-Jin Cho

**Affiliations:** 1Department of Pediatrics, School of Medicine, Kyungpook National University, Daegu 41566, Korea; ponyks1004@naver.com (D.-E.R.); lovecello623@gmail.com (J.-E.K.); 2Department of Pediatrics, Busan Baik Hospital, Inje University College of Medicine, Busan 50834, Korea; 3Department of Pediatrics, Chonnam National University Children’s Hospital, Gwangju 61469, Korea; hjina01@naver.com (H.N.); realalice@hanmail.net (I.C.); 4Department of Pediatrics, School of Medicine, Chonnam National University, Gwangju 61469, Korea

**Keywords:** COVID-19, vaccine, chest pain, myocarditis

## Abstract

As adolescents started to be vaccinated against coronavirus disease 2019 (COVID-19), suspected myocarditis and pericarditis related to the vaccine were reported in adolescents. According to the Korea Disease Control and Prevention Agency (KDCA), 2,796,270 persons aged 12–18 years were fully vaccinated by December 8. Among these, 9223 adverse events were reported (0.33%). We aimed to elucidate the clinical courses and short-term outcomes for adolescents aged 12–18 with cardiac symptoms and suspected myo- or peri-carditis related to COVID-19 vaccination in South Korea. Methods: We retrospectively collected data on patients ≤ 18 years of age who had suspected myocarditis or pericarditis within 30 days of COVID-19 vaccination, from July 2021 to January 2022. Results: We reported on 40 adolescents in different South Korean provinces at two centers. Twenty-six cases (65%) were male, and the median age was 16 years (range, 13–18; IQR 14.5–17). Twenty-five cases (62.5%) occurred at the first dose, and fifteen (37.5%) occurred after the second dose. Symptoms started at a median of 2 days (range 0–29 days; IQR 1–5 days) after vaccination. The patients were treated with nonsteroidal anti-inflammatory drugs (77.5%), intravenous immunoglobulin (2.5%), glucocorticoids (20%), colchicine (5%), or no therapy (15%). Five patients (12.5%) required intensive care unit admission; one patient needed inotropic/vasoactive support. No patients required extracorporeal membrane oxygenation or died. The median hospital stay was one day (range 0–8 days; IQR 0–2 days). Twenty-one patients (52.5%) had an abnormal electrocardiogram; among these, seven patients had an elevated ST segment, six patients (15%) had decreased ejection fraction (<55%), and LV function was completely recovered in all of them. Conclusions: Most cases of suspected myocarditis after COVID-19 vaccination in adolescents ≤ 18 years had mild symptoms and clinical courses, as well as a complete recovery. Further studies are needed to evaluate long-term outcomes.

## 1. Introduction

Since the end of 2019, SARS-CoV-2 has emerged as a new, potentially dangerous and concerning infectious disease, which eventually evolved into a global pandemic [1,2]. After intense research efforts, vaccines against SARS-CoV-2 were rapidly developed to protect global health, and vaccination soon began according to each country’s policies. Initially, vaccines were administered to adults. Meanwhile, some cases of myocarditis were reported in young adults after COVID-19 vaccination in Israel and the U.S. [3,4]. However, as the safety of the COVID-19 vaccine was reassuring in adults, administering COVID-19 vaccines to adolescents and children became standard practice in many countries [5]. 

In South Korea, the COVID-19 vaccination campaign for adolescents began in July 2021, when authorization was given to administer the COVID-19 vaccine to those as young as 12–17 years of age. Only the Pfizer-BioNTech messenger RNA-based (mRNA) COVID-19 vaccine was approved for adolescents in South Korea [6]. As more adolescents were administered with the COVID-19 vaccine, concerns of developing cardiac symptoms, with suspected myocarditis and/or pericarditis, began to increase. Therefore, there is a need to analyze the characteristics and clinical courses of the COVID-19 vaccine to learn more about its adverse events in Korean adolescents. Herein, we aim to elucidate the clinical courses and short-term outcomes of suspected myocarditis and pericarditis related to COVID-19 vaccination in South Korean adolescents from 12 to 18 years of age.

## 2. Materials and Methods

We retrospectively collected data on patients ≤ 18 years of age who presented with cardiac symptoms and had suspected myocarditis or pericarditis within 30 days of COVID-19 vaccination, from July 2021 to January 2022, from two national children’s hospitals located in different provinces in South Korea. We included patients with chest pain, pressure, chest discomfort, dyspnea, shortness of breath, palpitation or syncope after COVID-19 vaccination. We excluded patients with cardiac symptoms unrelated to COVID-19 vaccination or those that occurred 30 days after the COVID-19 vaccine. The study was approved by the institutional review boards of each center, and waivers of consent were granted (CNUH-2022-012 and KNUCH 2022-02-037). We collected demographic, clinical, laboratory, chest X-ray, electrocardiogram (ECG), echocardiogram (ECHO), and short-term outcomes. All of the hospitalized patients were laboratory-confirmed as negative for SARS-CoV-2. Left ventricular systolic function was deemed normal if left ventricular ejection fraction (LVEF) was >55%; decreased systolic function was defined as mildly impaired if LVEF was 41–55% and moderately or severely impaired if LVEF was 40% or less. LVEF was measured by either Teicholz method or Simpson monoplane four-chamber methods. E/A ratio, E/E’ ratio, and tricuspid annular plane systolic excursion (TAPSE) were additionally analyzed. The decision and timing to obtain imaging studies and management were at the discretion of each local team. Probable and confirmed cases of COVID-19-vaccine-associated myocarditis were defined according to CDC case definition [7]. 

### Statistical Analyses

Descriptive statistics include percentages for discrete variables and median values, with range and interquartile range (IQR) for continuous variables. All statistical analyses were performed using R version 4.1.0.

## 3. Results

### 3.1. Patients’ Clinical Characteristics

As of 15 January 2022, we have collected data from 56 adolescents at two centers with chest pain and clinically suspected myocarditis related to the COVID-19 vaccine. Sixteen patients did not undergo echocardiographic evaluation because the patients had milder symptoms, with normal ECG and laboratory findings. Out of the remaining 40 patients, as described in Table 1, 26 cases (65%) were male, and the median age was 16 years (range 13–18; IQR 14.5–17). Twenty-five (62.5%) occurred at the first dose, and fifteen (37.5%) occurred after the second dose. Symptoms started at a median of 2 days (range 0–29 days; IQR 1–5 days) after vaccination. The hospitalized patients were treated with nonsteroidal anti-inflammatory drugs (77.5%), intravenous immunoglobulin (2.5%), glucocorticoids (20%), colchicine (5.0%), or no therapy (15%). Five patients (12.5%) required pediatric intensive care unit (PICU) admission; one patient needed inotropic/vasoactive support for transient hypotension. No patients required extracorporeal membrane oxygenation or died. The median hospital stay was 1 day (range 0–8 days; IQR 0–2 days). 

### 3.2. Laboratory Findings 

The median (range; IQR) of laboratory findings in all 40 cohorts is described in Table 2. Five patients (12.5%) had elevated troponin I level, and five patients (12.5%) showed abnormally high NT-Pro BNP levels. 

### 3.3. Electrocardiographic and Echocardiographic Data

Twenty-one patients (52.5%) had an abnormal electrocardiogram; among these, seven patients had elevated ST segment. A premature ventricular contraction was seen in two patients while evaluating chest pain. Both patients did not have a previous history of arrythmia and did not have any associated symptoms during admission and since the events. Six patients (15%) had decreased ejection fraction (<55%); no patient had EF ≤ 30% (Table 3). Eight patients (20%) had transient mitral valve insufficiency. Three patients (7.5%) had a minimal amount of pericardial effusion and none of them required intervention. 

### 3.4. Characteristics of Adolescents with Suspected Myocarditis and/or ICU Admission 

Table 4 describes the patient characteristics of adolescents who had chest pain and suspected myocarditis related to COVID-19 vaccination and/or who required ICU admission. Of those eight patients, five had transient LV dysfunction and required IV steroids and/or IVIG for treatment. All patients survived until discharge with completely recovered LV function. Case six (a 14-year-old girl) was vaccinated on 10 November 2021 and, 3 days later, started to show symptoms of nausea and vomiting. She was treated under the diagnosis of gastroenteritis with no improvement in symptoms, and then transferred to one of the participating centers. The ECG revealed wide QRS tachycardia with a right bundle branch block pattern; however, no structural abnormalities were revealed. Under the diagnosis of idiopathic left ventricular tachycardia (ILVT), intravenous verapamil was given and the ILVT was terminated. No recurrent attack has been noted since this initial attack.

## 4. Discussion

Although myocarditis and pericarditis were not reported in COVID-19 clinical trials, only post-marketing surveillance can detect rare adverse events. In fact, some reports of suspected cases following vaccination in the general population have appeared. In this case series, we reported on 56 adolescents ≤ 18 years of age who had cardiac symptoms and clinically suspected myocarditis related to COVID-19 vaccination. Symptoms and clinical findings developed within a week after vaccination (IQR: 1–4 days), and about 68% occurred after the first dose of the vaccine. The most frequent changes in ECG were ST-T wave changes (12.5%). Approximately 15% of affected adolescents showed reduced LVEF on the echocardiogram at presentation, which was normalized in all patients. No RV systolic dysfunction was observed in this cohort. Less than 10% of patients were admitted to PICU; the PICU stay was short, and the requirement of inotropic/vasoactive agents was low. No death occurred in our case series, and no patient required extracorporeal membrane oxygenation. All patients with mildly impaired LVEF completely recovered LV function. This was fundamentally different from the typical myocarditis in childhood, which leaves a residuum in 30–60% of cases and sometimes even with further progression [8,9]. 

In South Korea, as of 8 December 2021, 2,796,270 persons aged 12–17 years have been fully vaccinated, according to the Korea Disease Control and Prevention Agency (KDCA) [6]. Considering the 56 cases we were able to collect, the frequency of chest pain post-vaccination with the Pfizer-BioNTech vaccine in adolescents in South Korea appears to be approximately two cases for every 100,000 people (i.e., considering two doses for each person, approximately one case for every 200,000 vaccine administrations). However, we excluded 16 patients who had normal ECG findings and normal laboratory findings. These 16 patients were more likely to have myalgia rather than myocarditis. Eventually, adverse event reports followed since the vaccination was started for adolescents, as other countries reported. Among 519,005 vaccinated adolescents, 9223 cases reported adverse events (0.33%), and this occupied 72% of all reported adverse events in all age groups (0.46%) [6]. Although over 95% of reported adverse events were mild, 27 cases of suspected myocarditis/pericarditis in persons aged 12–17 years were reported in Korea. Among these, five were confirmed (0.2 cases/100,000 vaccination), all of which were entirely resolved. 

The Center for Disease Control and Prevention (CDC) recommends that everyone aged 5 years and older receives a COVID-19 vaccine, and only the Pfizer-BioNTech (BNT) 162b2 mRNA COVID-19 vaccine is authorized for children and adolescents over 5 years old [10]. In addition, everyone aged 12 years and older should receive a COVID-19 booster shot at least 5 months after completing the primary COVID-19 vaccination series [11]. As it is almost time for previously vaccinated adolescents and their families to decide whether to have their booster shots or not, information on the clinical courses of those who suffered from adverse events may be useful. 

Vaccine-associated serious adverse events, such as myocarditis, are the greatest concern for adolescents and their families, even if the clinical courses are mild. Unlike the clinical course of multisystem inflammatory syndrome in children (MIS-C) [12,13,14], in adolescents with suspected myocarditis related to COVID-19 vaccination, data suggest that myocarditis has a mild hospital course, with a quick clinical recovery and excellent middle-term outcomes, as well as a complete resolution without sequelae in adolescents [15]. Furthermore, myocarditis has been associated with other vaccines, such as smallpox [16] and influenza [17,18]. Interestingly, a meta-analysis compared COVID-19-vaccine-related myopericarditis to non-COVID-19-vaccine-related myopericarditis [19]. This meta-analysis reported that the risk of myopericarditis amongst those receiving COVID-19 vaccination was not different from non-COVID-19 vaccines for the general population. Of note, a recent systematic review, including 34 patients under 20 years of age with myocarditis, presented 100% recovery and 0% death from COVID-19-vaccine-related myocarditis [20]. Our case series adds further support to these results.

## 5. Limitations

This study has some limitations. First, the included number of patients was small, and the study was retrospectively conducted. Second, although our cohort was from the only children’s hospital in each province, this cohort may not represent all adolescents across the country. Third, it was challenging to obtain an endomyocardial biopsy or cardiac MRI to confirm myocarditis under pandemic circumstances, as resources and personnel were limited, and no patient underwent an endomyocardial biopsy or cardiac MRI. Fourth, although we elucidated middle-term outcomes for Korean adolescents, this study does not include long-term outcomes, which need to be studied in the near future.

## 6. Conclusions

In conclusion, most adolescents presenting with chest pain and suspected myocarditis after COVID-19 had a mild clinical course and presented within 1 week after vaccination. About 15% of adolescents presented with chest pain and suspected myocarditis and had depressed LV systolic function, but the LVEF was normalized in all adolescents. Further studies are needed to evaluate long-term outcomes. 

## Figures and Tables

**Table 1 children-09-00693-t001:** Vaccination.

Clinical Characteristics	Value
Total number of adolescents	40
Age, year (range; IQR)	16 (13–18; 14.5–17)
Male sex, *n* (%)	26 (65)
Asian, *n* (%)	40 (100)
Dose of vaccine with symptoms, *n* (%)	
1st dose, *n* (%)	25 (62.5)
2nd dose, *n* (%)	15 (37.5)
Days from vaccination to symptom occurrence (range; IQR)	2 (0–29; 1–5)
Cardiac symptoms, *n* (%)	
Chest pain, pressure, chest discomfort	40 (100)
Dyspnea, shortness of breath	7 (17.5)
Palpitations	5 (12.5)
Syncope	1 (2.5)
Treatment	
NSAIDS, *n* (%)	31 (77.5)
IVIG, *n* (%)	1 (2.5)
Glucocorticoids, *n* (%)	8 (20.0)
Colchicine, *n* (%)	2 (5.0)
Only supportive	6 (15.0)
Hospital stay, days (range; IQR)	1 (0–8; 0–3)
ICU admission, *n* (%)	5 (12.5)
PICU stay, days (range; IQR)	3 (1–6; 1–3.75)
Adolescents requiring inotropics/vasoactive agents, *n* (%)	1 (2.5)
Adolescents requiring ECMO, *n* (%)	0
Mortality, *n* (%)	0

COVID-19, coronavirus disease 2019; ECMO, extracorporeal membrane oxygenation; IVIG, intravenous immunoglobulin; IQR, interquartile range; NSAIDS, nonsteroidal anti-inflammatory drugs; PICU, pediatric intensive care unit.

**Table 2 children-09-00693-t002:** Laboratory findings of an adolescent with chest pain and suspected myocarditis related to COVID-19 vaccination.

Laboratory Findings	Value, Median (Range; IQR)
WBC, G/L (*n* = 40)	7100 (4100–12,900; 6015–8600)
Neutrophil, G/L (*n* = 40)	4010 (1920–10,400; 2860–5260)
Lymphocyte, G/L (*n* = 40)	2150 (810–3320; 1740–2660)
Monocyte, G/L (*n* = 40)	480 (250–1540; 407–660)
Hemoglobin, g/L (*n* = 40)	144 (111–168; 136–152)
Platelet, G/L (*n* = 40)	266 (181–420; 221.7–295.2)
C-reactive protein, mg/L (*n* = 40)	9.4 (0.2–140; 1.6–50)
NT-pro BNP, pg/mL (*n* = 40) (Reference normal <125 pg/mL)	24.3 (8–1591; 13.5–47.0)
Troponin I, ng/mL (*n* = 40) (Reference normal value ≤ 0.014 ng/mL)	0.01 (0.003–4.8; 0.003–0.01)
Troponin T, ng/mL (*n* = 16) (Reference normal value ≤ 0.014 ng/mL)	0.003 (0.003–1.03; 0.003–0.52)
Creatine kinase, µg/L (*n* = 40)	71.0 (0.2–659; 1.02–100.7)
CK-MB, µg/L (*n* = 8)	4.8 (1.3–27.4; 2.1–24.6)

NT-pro BNP, N-terminal prohormone B-type natriuretic peptide; IQR, interquartile range; WBC, white blood cells.

**Table 3 children-09-00693-t003:** Electrocardiographic and echocardiographic data of adolescents with chest pain and suspected myocarditis related to COVID-19 vaccination.

Variables	Value
ECG tested, *n* (%)	40 (100)
Abnormal	21 (52.5)
Normal	19 (47.5)
ECG findings, overlaps allowed	
ST and/or T wave changes	7 (17.5)
Ventricular tachycardia	1 (2.5)
Low-voltage QRS	0
Premature ventricular contractions	2 (5.0)
Premature atrial contractions	1 (2.5)
Sinus tachycardia	4 (10.0)
Sinus bradycardia	5 (12.5)
ECHO tested, *n* (%)	40 (100)
Left ventricular function, *n* (%)	40 (100)
EF (%), median (range; IQR)	65 (range 40–88; IQR 59–71)
Normal EF, > 55%, *n* (%)	34 (85)
Mildly impaired EF, 41–55%, *n* (%)	6 (15)
E/A (*n* = 39)	1.83 (range 1.07–3.87; IQR 1.45–2.12)
Abnormal E/A, *n* (%)	0
E/E’ (*n* = 40)	7.59 (range 4.22–12.07; IQR 6.87–8.42)
Abnormal E/E’, *n* (%)	0
Right ventricular function, *n* (%)	22 (57.5)
TAPSE (absolute value, mm), median (range; IQR)	23 (range 17–33; IQR 21–26.5)
Abnormal TAPSE, *n* (%)	0
Mitral valve insufficiency, *n* (%)	8 (20)
Pericardial effusion, *n* (%)	3 (7.5)

ECG, electrocardiogram; ECHO, echocardiogram; EF, ejection fraction; IQR, interquartile range; TAPSE, tricuspid annular plane systolic excursion.

**Table 4 children-09-00693-t004:** Patient characteristics of adolescents who had suspected myocarditis related to COVID-19 vaccination and/or who required ICU admission.

Cases	Age (Years)	Sex	Vaccine Dose	PICU Stay (Days)	Hospital Stay (Days)	Days from Vaccine to Symptom Onset	Troponin I or T	NT-Pro BNP	ECG Abnormality	LVEF (%)	Pericardial Effusion	Treatment
1	17	M	2nd	3	7	1	0.453	312	ST elevation	40	No	IV steroid
2	17	M	2nd	1	7	3	0.175	92	Sinus bradycardia	51	No	NSAIDS IV steroids
3	16	M	2nd	0	7	1	1.030	189	ST elevation	59	Yes, minimal	IV steroid Colchicine
4	13	M	2nd	0	6	3	0.518	177	NSR	51	No	NSAIDS IV steroids
5	17	M	1st	0	2	12	4.8	407.7	ST elevation	65	No	NSAIDS
6	14	F	1st	6	8	3	0.036	1591	Ventricular Tachycardia	44	No	NSAIDS IV steroids Verapamil
7	17	F	1st	1	3	1	0.01	21.8	NSR	75	No	No therapy
8	16	M	1st	3	6	5	0.013	11.8	NSR	54	No	NSAIDS IV steroids IVIG

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
