# Peer review of "Chest Pain and Suspected Myocarditis Related to COVID-19 Vaccination in Adolescents—A Case Series"

_children, 2022, doi:10.3390/children9050693_

Round 1

Reviewer 1 Report

Dear authors,

Congratulatons for putitng together an interesting report of over 50 cases of suspected myocarditis following COVID-19 vaccination with mRNA vaccine. The topic is of considerable interest in the pandemic, and these data can be used for decision-making regarding vaccination of children and associated risks for the overall population. Although the report is interesting, I have some points to be addressed, as you can see below:

1) Intro: considering the nature of the manuscript (case series), the Intro can be focused and shortened, without losing the key message. Please focus on mRNA vaccine and the reports and concerns regarding the development of complications and myocarditis.

2) About the echo data, please report the number of patients with low ejection fraction (<30%) and intermediate EF (31-40%), as recommended by guidelines.

3) Also about the echo, please report other findings related to myocarditis, especially those related to the inflamamtory proccess (hyperrefringency, abnormal wall motion, altered wall thickness) and other associated abnormalities (pericardial effusion etc.).

4) Please extend the paragraph describing the echo findinds, including other relevant observations, as those suggested in item 3.

4) Do you have MRI data for these patients? If so, please report, in addition to the echo and ECG data. If not, please report as a limitation of the study.

5) Discussion: are there any concerns about the final diagnoses of such patients? I mean: do doubts remain about the real diagnosis of vaccine-associated myocarditis? If so, what could be the alternate diagnoses? Please improve the discussion surrounding these points.

6) Is possible, please look at the demographic, clinical and ECG/echo variables associated with unfavorable outcome (ICU admission, for example). This would add to the existing information about risk-stratification for the development of severe heart disease following vaccination with mRNA.

Author Response

Thank you very much for your time and we all appreciate your input. Please find point by point responses.

Congratulatons for putitng together an interesting report of over 50 cases of suspected myocarditis following COVID-19 vaccination with mRNA vaccine. The topic is of considerable interest in the pandemic, and these data can be used for decision-making regarding vaccination of children and associated risks for the overall population. Although the report is interesting, I have some points to be addressed, as you can see below:

1) Intro: considering the nature of the manuscript (case series), the Intro can be focused and shortened, without losing the key message. Please focus on mRNA vaccine and the reports and concerns regarding the development of complications and myocarditis.

: We agree that the introduction need to be focused and shortened. We have moved following paragraph to discussion.  “Adverse events inevitably followed since the vaccination for adolescents in South Korea. Among the vaccinated adolescents, 9,223 cases were reported adverse events (0.33%), and this occupies 72% of all reported adverse events in all age groups (0.46%).6 Although over 95% of reported adverse events were mild symptoms, about 27 cases were reported suspected myocarditis/pericarditis in persons aged 12-17 years in Korea. Among these, five were confirmed (0.2cases/100,000 vaccination), all of which were resolved entirely. Persons aged 18 years, 908,101 were vaccinated, 41 reported adverse events for suspected myocarditis/pericarditis, and 23 cases (2.53 cases/100,000 vaccination) were confirmed, and all were resolved entirely.6” 

2) About the echo data, please report the number of patients with low ejection fraction (<30%) and intermediate EF (31-40%), as recommended by guidelines.

: Thank you for pointing out this issue. In table 3, we added intermediate EF and Low EF line. Although we had 0 patients with EF <=30%, we agree with you that showing the result is also meaningful.

3) Also about the echo, please report other findings related to myocarditis, especially those related to the inflamamtory proccess (hyperrefringency, abnormal wall motion, altered wall thickness) and other associated abnormalities (pericardial effusion etc.).

: Thank you for your suggestion. We mostly collected data of functional assessment including EF, E/A, E/E’, GLS, TAPSE, S’ and presented from these data. We presented the number of patients with pericardial effusion and mitral valve insufficiency.

4) Please extend the paragraph describing the echo findinds, including other relevant observations, as those suggested in item 3.

:We have added those collected data to the table and to the main body.

Six patients (15%) had decreased ejection fraction (< 55%); median EF was 50% (range 40-52; IQR44-51) and no patient had EF 30 %. No patient had LV diastolic dysfunction or RV dysfunction by echocardiography (Table 3). Eight patients (20%) had transient mitral valve insufficiency. Three patients had minimal amount of pericardial effusion and none of them required intervention.”

4) Do you have MRI data for these patients? If so, please report, in addition to the echo and ECG data. If not, please report as a limitation of the study.

: Unfortunately, during the pandemic circumstances, it was difficult to perform myocardial biopsy and cardiac MRI in all patients. We reported this situation as a limitation of this study. Thank you.

“Second, it was challenging to obtain a myocardial biopsy or cardiac MRI during active pandemic circumstances to confirm myocarditis as the resources and personnel were limited.”

5) Discussion: are there any concerns about the final diagnoses of such patients? I mean: do doubts remain about the real diagnosis of vaccine-associated myocarditis? If so, what could be the alternate diagnoses? Please improve the discussion surrounding these points.

: Thank you for your suggestion. Most adolescent with chest pain was not myo/pericarditis and resolved quickly. This was thought to be psychotic fear of new vaccination among parents and the children. We added this in the limination section that most were not diagnosed of vaccine associated myocarditis.

the final diagnoses of most adolescents with chest pain were not myo/pericarditis in this cohort, however this also reassures the concerns of new vaccination in youth”.

6) Is possible, please look at the demographic, clinical and ECG/echo variables associated with unfavorable outcome (ICU admission, for example). This would add to the existing information about risk-stratification for the development of severe heart disease following vaccination with mRNA.

: We agree with you that further analyses will elaborate better however, the number of adolescents with unfavorable outcome were too small (5 ICU admitted patients, 1 inotropic required patient, none ECMO patient) it was hard to draw conclusions. However, we will further collect data and try to analyse further as you suggested. Thank you for your suggestion.

: Thank you again for reviewing our manuscript. We have revised according to your comments and tried to improve the discussion section. Please do let us know if we need to revise further.

The English will be further edited by a native speaker after the revision. Thank you.

Reviewer 2 Report

The study is good study to analyse the cardiac side effects of mRNA vaccine.  Some questions if can be clarified will add value.

(1) The criteria adopted for diagnosis of myo-percarditis is not defined in the study.

(2) The pick up was only apparently by chest pain which may not be appropriate. There can be other symptoms also ( though less common)

(3) In a cohort of 56 patients , the diagnostic work up not done in all. Maximum number of patient included 55 in number. How myopericaritis confirmed in all and the missed one patient not clear.

(4) Why echocardiogram done in 40 patients only ? 16 patients did not have a prominent evaluation thereby limiting the scope of diagnosis.

(5) even blood test not uniformly done in all. The methodology should specify the criteria for diagnosis and limitation in this regard.

(6) The possible reason for the side effect not discussed as the comparative literature on adolescent myocarditis post covid vaccination. There are comparative studies with inactivated vaccines.

The discussion section can possibly elaborate more available literature.

Some prominent studies which possibly enhance the discussion

[1] Krug A, Stevenson J, Høeg TB. BNT162b2 Vaccine-Associated Myo/Pericarditis in Adolescents: A Stratified Risk-Benefit Analysis. Eur J Clin Invest. 2022 Feb 14:e13759. doi: 10.1111/eci.13759. Epub ahead of print. PMID: 35156705

[2] Lai FTT, Li X, Peng K, Huang L, Ip P, Tong X, Chui CSL, Wan EYF, Wong CKH, Chan EWY, Siu DCW, Wong ICK. Carditis After COVID-19 Vaccination With a Messenger RNA Vaccine and an Inactivated Virus Vaccine : A Case-Control Study. Ann Intern Med. 2022 Jan 25:M21-3700. doi: 10.7326/M21-3700. Epub ahead of print. PMID: 35073155; PMCID: PMC8814917.

[3] Witberg G, Barda N, Hoss S, Richter I, Wiessman M, Aviv Y, Grinberg T, Auster O, Dagan N, Balicer RD, Kornowski R. Myocarditis after Covid-19 Vaccination in a Large Health Care Organization. N Engl J Med. 2021 Dec 2;385(23):2132-2139. doi: 10.1056/NEJMoa2110737. Epub 2021 Oct 6. PMID: 34614329; PMCID: PMC8531986

[4] Perez Y, Levy ER, Joshi AY, Virk A, Rodriguez-Porcel M, Johnson M, Roellinger D, Vanichkachorn G, Huskins WC, Swift MD. Myocarditis Following COVID-19 mRNA Vaccine: A Case Series and Incidence Rate Determination. Clin Infect Dis. 2021 Nov 3:ciab926. doi: 10.1093/cid/ciab926. Epub ahead of print. PMID: 34734240; PMCID: PMC8767838.

[5] COVID-19 Vaccine Safety Updates

Vaccines and Related Biological Products Advisory Committee (VRBPAC)

June 10, 2021

Tom Shimabukuro, MD, MPH, MBA

Vaccine Safety Team
CDC COVID-19 Vaccine Task Force

Author Response

Reviewer 2

Thank you very much for your time and we all appreciate your input. Please find point by point responses.

The study is good study to analyse the cardiac side effects of mRNA vaccine.  Some questions if can be clarified will add value.

(1) The criteria adopted for diagnosis of myo-percarditis is not defined in the study.

: Thank you for pointing out this important issue. In the method section we added a sentence with CDC definition of probable and confirmed cases for COVID-19 vaccine associated myocarditis.

“Probable and confirmed case of COVID-19 vaccine associated myocarditis were defined according to CDC case definition. (Gargano JW, Wallace M, Hadler SC, Langley G, Su JR, Oster ME, Broder KR, Gee J, Weintraub E, Shimabukuro T, et al.. Use of mRNA COVID-19 vaccine after reports of myocarditis among vaccine recipients: update from the Advisory Committee on Immunization Practices – United States, June 2021.MMWR Morb Mortal Wkly Rep. 2021; 70:977–982. doi: 10.15585/mmwr.mm7027e2)”

(2) The pick up was only apparently by chest pain which may not be appropriate. There can be other symptoms also ( though less common)

: We agree. Although we simplified it to chest pain, some expressed as pressure or chest discomfort. We changed our method section according to your comment.

“We included patients with chest pain, pressure, chest discomfort, dyspnea, shortness of breath, palpitation or syncope after COVID-19 vaccine. We excluded patients with cardiac symptoms unrelated to the COVID-19 vaccine or that occurred 30 days after the COVID-19 vaccine.”

(3) In a cohort of 56 patients , the diagnostic work up not done in all. Maximum number of patient included 55 in number. How myopericaritis confirmed in all and the missed one patient not clear.

: We really appreciate you for being critical with the numbers. We have reviewed the raw data and statistics and it turned out to be a type error. All patients underwent CBC with routine chemistry and ECG but not all underwent cardiac related laboratory tests and echocardiogram. We do understand your concern that how myopericarditis was confirmed, but the authors wanted to reassure with this study that not all symptomatic patients were myopericarditis. Some symptomatic patients might have slight decrease of EF or changes of ECG or elevated NT proBNP/troponin i/troponin T but the number of probable myocarditis related to COVID-19 vaccine was low in this study. We hope this answers your concern. Thank you.

(4) Why echocardiogram done in 40 patients only ? 16 patients did not have a prominent evaluation thereby limiting the scope of diagnosis.

: This is definitely our limitation of this study. Some of patients with mild symptom with no ECG changes and no laboratory abnormalities did not undergo echocardiographic evaluation in pandemic circumstances with limited personnel. We agree to your concern and added this to our limitation section.

(5) even blood test not uniformly done in all. The methodology should specify the criteria for diagnosis and limitation in this regard.

: Yes. We do agree that the blood tests were not uniformly done in all. As this study was retrospective review of clinical data, we added to our limitation.

“. First, the included number of patients were small and the study was conducted retrospectively; the tests were not uniformly performed in the results.”

(6) The possible reason for the side effect not discussed as the comparative literature on adolescent myocarditis post covid vaccination. There are comparative studies with inactivated vaccines.

 The discussion section can possibly elaborate more available literature.

: Thank you very much for your thoughtful suggestions. As we cannot fully diagnose myocarditis and cannot rule out other possible reasons we added a sentence to our limitation

“the final diagnoses of most adolescents with chest pain were not myo/pericarditis in this cohort, however this also reassures the concerns of new vaccination in youth”

Some prominent studies which possibly enhance the discussion

: Thank you very much for your suggestion. We have reviewed and added 2 more references to our main body.

[1] Krug A, Stevenson J, Høeg TB. BNT162b2 Vaccine-Associated Myo/Pericarditis in Adolescents: A Stratified Risk-Benefit Analysis. Eur J Clin Invest. 2022 Feb 14:e13759. doi: 10.1111/eci.13759. Epub ahead of print. PMID: 35156705 

[2] Lai FTT, Li X, Peng K, Huang L, Ip P, Tong X, Chui CSL, Wan EYF, Wong CKH, Chan EWY, Siu DCW, Wong ICK. Carditis After COVID-19 Vaccination With a Messenger RNA Vaccine and an Inactivated Virus Vaccine : A Case-Control Study. Ann Intern Med. 2022 Jan 25:M21-3700. doi: 10.7326/M21-3700. Epub ahead of print. PMID: 35073155; PMCID: PMC8814917. 

[3] Witberg G, Barda N, Hoss S, Richter I, Wiessman M, Aviv Y, Grinberg T, Auster O, Dagan N, Balicer RD, Kornowski R. Myocarditis after Covid-19 Vaccination in a Large Health Care Organization. N Engl J Med. 2021 Dec 2;385(23):2132-2139. doi: 10.1056/NEJMoa2110737. Epub 2021 Oct 6. PMID: 34614329; PMCID: PMC8531986 

[4] Perez Y, Levy ER, Joshi AY, Virk A, Rodriguez-Porcel M, Johnson M, Roellinger D, Vanichkachorn G, Huskins WC, Swift MD. Myocarditis Following COVID-19 mRNA Vaccine: A Case Series and Incidence Rate Determination. Clin Infect Dis. 2021 Nov 3:ciab926. doi: 10.1093/cid/ciab926. Epub ahead of print. PMID: 34734240; PMCID: PMC8767838. 

[5] COVID-19 Vaccine Safety Updates

Vaccines and Related Biological Products Advisory Committee (VRBPAC)

June 10, 2021

Tom Shimabukuro, MD, MPH, MBAVaccine Safety Team
CDC COVID-19 Vaccine Task Force

: Thank you again for reviewing our manuscript. We have revised according to your comments and tried to improve the discussion section. Please do let us know if we need to revise further.

The English will be further edited by a native speaker after the revision. Thank you.

Reviewer 3 Report

I reviewed the manuscript titled" Chest pain and Suspected Myocarditis related to COVID-19 vaccination in Adolescents". There are some remarks:

1- Some English errors should be revised. 

2- In all parts of the articles, it is mentioned that 56 patients are suspected to myocarditis, but in table 4, only 8 patients are presented in detail. It is better to present more details on the suspected patients as a supplementary table.

3- There are not enough explanations about the diagnostic criteria and definition of myocarditis in the methods section. The authors should clarify how did they suspect this disorder and how did they diagnose that in the patients.

4- There are limited data on the total sample size that had been vaccinated. More information on the total vaccinated population is better to be reported.

5- The discussion needs improvement. Many similar studies have been published, and it is better to discuss and compare the findings. (e.g. https://doi.org/10.1155/2022/2438913)

Author Response

Reviewer 3

Thank you very much for your time and we all appreciate your input. Please find point by point responses.

I reviewed the manuscript titled" Chest pain and Suspected Myocarditis related to COVID-19 vaccination in Adolescents". There are some remarks:

1- Some English errors should be revised. 

: We will revise the English by native English speaker. Thank you.

2- In all parts of the articles, it is mentioned that 56 patients are suspected to myocarditis, but in table 4, only 8 patients are presented in detail. It is better to present more details on the suspected patients as a supplementary table.

: Thank you for your comments. As the 56 patients in a table might look too busy, we decided to present those who were suspected myocarditis related to COVID-19 vaccination and/or who required ICU admission. However, we agree the more details would be better to present.

3- There are not enough explanations about the diagnostic criteria and definition of myocarditis in the methods section. The authors should clarify how did they suspect this disorder and how did they diagnose that in the patients.

: Thank you for pointing out this important issue. In the method section we added a sentence with CDC definition of probable and confirmed cases for COVID-19 vaccine associated myocarditis.

“Probable and confirmed case of COVID-19 vaccine associated myocarditis were defined according to CDC case definition. (Gargano JW, Wallace M, Hadler SC, Langley G, Su JR, Oster ME, Broder KR, Gee J, Weintraub E, Shimabukuro T, et al.. Use of mRNA COVID-19 vaccine after reports of myocarditis among vaccine recipients: update from the Advisory Committee on Immunization Practices – United States, June 2021.MMWR Morb Mortal Wkly Rep. 2021; 70:977–982. doi: 10.15585/mmwr.mm7027e2)”

4- There are limited data on the total sample size that had been vaccinated. More information on the total vaccinated population is better to be reported.

: Thank you for your comments. We have added a paragraph in the discussion section regarding the total vaccinated adolescent near the time of our study.

“In South Korea, as of December 8, 2021, 2,796,270 persons aged 12-17 years completed vaccination, according to Korea Disease Control and Prevention Agency (KDCA).6 Eventually, adverse event reports followed since the vaccination was started for adolescents, as other countries reported. Among 519,005 vaccinated adolescents, 9,223 cases were reported adverse events (0.33%), and this occupies 72% of all reported adverse events in all age groups (0.46%) as of .6 Although over 95% of reported adverse events were mild symptoms, about 27 cases were reported suspected myocarditis/pericarditis in persons aged 12-17 years in Korea. Among these, five were confirmed (0.2cases/100,000 vaccination), all of which were resolved entirely.”

5- The discussion needs improvement. Many similar studies have been published, and it is better to discuss and compare the findings. (e.g. https://doi.org/10.1155/2022/2438913)

: Thank you again for reviewing our manuscript. We have revised according to your comments and tried to improve the discussion section. Please do let us know if we need to revise further.

The English will be further edited by a native speaker after the revision. Thank you.

Round 2

Reviewer 1 Report

Dear authors. Thank you for addressing the points I've highlighted. I have no additional comments at this point.

Author Response

Thank you again for reviewing our manuscript.

Reviewer 3 Report

All of my comments have been sufficiently addressed. 

Author Response

(The authors gave the same response as above.)
